# Impact of tourism on sustainable development in BRI countries: The moderating role of institutional quality

**Huma Iftikhar**[1], **Chen Pinglu**[1]*, **Saif Ullah**[2], **Atta Ullah**[1]*

**1** School of Management, Huazhong University of Science and Technology (HUST), Wuhan, China, **2** ILMA University, Karachi, Pakistan

* attaullah142@gmail.com (AU); cplcmhust@hust.edu.cn (CP)

**Data Availability Statement:** All relevant data are within the manuscript and its Supporting Information files.

**Funding:** The author(s) received no specific funding for this work.

## Abstract

This study investigated the influence of tourism on sustainable development while considering institutional quality as a moderating variable. Moreover, exchange rate, urbanization, household consumption, per capita income and renewable energy per capita were also essential factors in determining sustainable development. The sample consists of 64 Belt and Road Initiative (BRI) countries from 2003–2018. The outcomes of the two-step system GMM confirmed the statistically significant and positive dynamic nature of sustainable development and its relationship with tourism and other determinants at a significance level of 1% for BRI countries. Institutional quality enhanced the 4.693% sustainability path to achieve the sustainable development goals (SDGs) agenda with regionally interconnected countries at significance level of 1%. Renewable energy per-capita and income per-capita played a significant and positive role, while the exchange rate, household consumption, and urbanization negatively influenced by hurting thd path of sustainable development. The current research findings have valuable contributions to academics as it offers novel insights about the 0. 351% influence of tourism on sustainable development at significance level of 1%, and it proposes valued suggestions to policymakers concerning tourism development strategies.

## 1. Introduction

In recent decades, the sustainable development' concept acquired massive consideration in the socio-economic and managerial literature. The perception of sustainable development refers to fulfil the current generation's needs by providing a quality of life without compromising future generations' requirements [1]. The United Nations (UN) General Assembly developed a worldwide agreement of SDGs in 2015 to drive the 2030 Agenda for sustainable development. It comprises all the UN member states to accomplish substantial sustainability identifying 169 targets and 17 goals before 2030 [2]. Amongst these 17 objectives, the 12th objective (SDG 12) demands activities to guarantee sustainable consumption and production designs comprising the executing tools for observing the influences of tourism on sustainable

**Competing interests:** The authors have declared that no competing interests exist

development [3]. For theimplementations of SDGs altogether states took some initiatives to achieve the goals of the 2030 agenda, while the tourism sector was the key sector in the 2030 Agenda [2].

Prior studies induce both positive and negative consequences of tourism toward growth. Few studies showed the negative consequences of tourism-growth as it causes unequal division of profits, loss of traditional culture, absence of native people in planning procedures, social conflict amongst hosts and guests, rise in land prices, higher costs for few services/goods, land speculation, numerous types of pollution, overcrowding, shortage of clean drinking water, production of litter, and other kinds of environmental degradation [3, 4]. Tourism is considered a development tool, but it degrades non-renewable and renewable resources [5]. So, the main criticism associated with tourism was the destruction of natural resources [6]. Additionally, tourism influences the lives of native people as it involves the construction of roads and hotels for tourists [2]. An increase in ecological and commercial challenges will harm economies, ecosystems, and human welfare [7].

Alternatively, previous literature has also shown the positive consequences of tourism-growth as local governments can improve public places' atmosphere and the value of cultural and natural resources through tourism development [3]. In the low and middle-income economies, tourism is associated with economic development as tourism activities create demographic stabilization, stimulation of agricultural development, rise in profitability of food stuffs, growth of local handicrafts, improvement of socio-economic wellbeing, the formation of new native enterprises, a rise of incomes, job opportunities, female employment, services up-gradation, improved quality of life, and advanced living standards for the residents. Furthermore, tourism motivates residents to preserve traditional, native, and natural heritage to improve the worth of tourists' experiences [4]. Generally, it also improves the quality of life for the inhabitants [8], declines outgoing migration, and improves sustainable development [4]. Fortunately, the positive impacts of tourism are more noticeable than adverse effects, as commercial influences are essential than socio-cultural and environmental effects. From an idealistic perspective, the tourism industry's presence in any region is beneficial because of its predictable, sustainable economic advantages. The tourism industry is progressively developing as a robust pillar of sustainable development all around the globe. Several developing and developed countries consider the tourism industry a source of income because it contributes to foreign exchange growth and employment creation. In the past two decades, tourism growth has helped in poverty alleviation, investment elevation, and the creation of employment [9]. Tourism support trade by building bridges among individuals from various cultures and regions. Tourism development contributes to GDP (Gross Domestic Product) by generating tax revenues and employment opportunities for the native people and contributes to infrastructure growth [10]. Tourism is renowned as one of the fastest developing sectors and vital to sustainable development worldwide [11]. Global tourist arrivals were 1.3 billion in 2017, and it was predicted to constantly rise every year by 3.3% until 2030 [3].

Institutions comprise formal/informal principles that define how individuals behave with one another. Various institutes endorse financial and sustainable development by creating confidence and collaboration, boosting investments, and discouraging free movement. Bad institutions bring economic recession, political instability, and corruption. So, institutional quality is a significant factor in differentiating sustainable development among countries [9]. Higher institutional quality promotes tourist inflows in destination countries [12, 13]. Good institutional quality has a constructive and economically noteworthy influence on sustainability [14–16]. So, the role of institutional quality is crucial amongst tourism and sustainable development.

In 2013, China proposed the BRI (Belt and Road Initiative) to connect Asia, Africa, and Europe to increase regional integration and boost sustainable development. The global scope of BRI is continuously rising as it covers more than 71 countries, which signifies almost 65% population globally [17]. BRI project has five key priorities: unhindered trade, policy synchronization, financial incorporation, infrastructure connectivity, and people-to-people connections [18]. These forecasts may return as essential productive forces for sustainable development and dynamic financial evolution for BRI countries [19]. As predicted, till 2049, BRI would be the biggest infrastructure project in human history, having a value of more than US$8 trillion [20]. Moreover, [21] stated that the mutual tourist flows could enhance the trade amongst countries. So, the role of tourism is essential for enhancing trade and sustainable development in the BRI region.

Previous researchers have investigated the factors that affect tourism, e.g. high-speed railway increases the approachability of ice-snow tourist destinations and contributing to the global tourism industry [22, 23]. Rural revitalization promotes rural tourism activities and contributes to sustainable rural development in developing countries [24]. While climate change is a critical factor that influences the tourism industry negatively. Humidity, sunshine duration and temperature affect hiking participation nonlinearly [25]. Climate-related changes also impact tourism as it reduce the duration of leaf colouration in Japan, which causes a delay in autumn foliage colouration change [26]. But, it has been almost ignored in the literature that tourist activities can influence sustainable development. Besides, [4] investigated how tourism development has affected the sustainable development for Romania only. That study was based on Romanian statistics data at the LAU-2 level, and this sustainable development data was poor as per the author because it does not permit contrasts to other states. Similarly, the role of institutional quality is crucial for both tourism and sustainable development, so institutional quality was added as a moderator as suggested by [27]. As compared to other countries, the BRI region has natural beauty so it is considered a preferred destination by worldwide tourists. Tourism is very important for the growth and revenues of BRI region as it comprises many low and middle-income nations. However, the influence of tourism on sustainable development was not explored for the BRI region, and institutional quality was not considered as a moderator between tourism and sustainable development in previous studies. Therefore, this research focuses on how tourism contributes to sustainable development with the moderating role of institutional quality by using country-level aggregated BRI countries' data from 2003 to 2018. This approach also facilitated the confirmation of whether urbanization, renewable energy per capita, exchange rate, per capita income, inflation, and household consumption would influence the sustainable development for the BRI region, which would be a vital contribution in the literature.

## 2. Literature review

Sustainable development is well-defined as fulfilling the current requirements without harming and compromising future needs [1, 28]. Also, [29] proposed the concept of 'triple bottom line" for the first time. This concept proposed that a sustainable society should have measurable environmental, economic, and social impactfull goals. Since, World War II, the tourism industry got attention when this idea was supported that the tourism activities would be a solution for local progress. Some tourist destinations have presented a growth of this activity since the 1970s because of low price strategies and mass tourism [30]. As tourism has been acknowledged as the economic force in several republics, the sustainable tourism model has developed over the last few years [31]. Categorical reference has been given to tourism in the 17 SDGs.

Specifically, SDGs' 8.9 and 12.7b have primary association with tourism [3]. This association is further explained based on theory and literature support.

## 2.1. Tourism and sustainable development

Tourism has been signified as one of the utmost vigorous industries globally as growth rates exceed all economic branches in recent decades. The tourism industry is one of the crucial drivers of sustainable development as created 292 million jobs (1 in 10 jobs in the world) and contributed US$7.6 trillion to the global economy (10.2% of the worldwide GDP) in 2016 [12].

Sustainable development is an essential strategy endorsed by various local governments and the United Nations World Tourism Organization (UNWTO). Further, [32] utilized Annual time-series data from 1990–2010 to determine sustainable development in Laos. This study proved that it is a combination of environmental protection, social development, and financial development.

Many native traders and tourism planners consider the tourism industry a driver of sustainable development [4]. Although it is considered that tourism development had only short-term effects as emigration process' reversal was detected when tourist activities started to flourish in demographically declining areas of Southern Europe [33], so the tourism industry is a vital contributor in the sustainable development of the region. In addition, [34] indicated that the tourism industry is associated with financial development for low and middle-income economies like Iran while [35] established a long-run equilibrium connection amongst tourism and financial development for Taiwan. Therefore, tourism was supposed as a sustainable 'cure-all' resolution for several problems of growing nations [36].

Few studies showed the negative impacts of tourism on sustainable development. As per [37], tourism boosts unstable Income and employment distributions in weak rural economies. Negative tourism influence has been associated with poor accessibility, low marketing techniques, a shortage of essential tourism and entrepreneurial expertise, and a shortage of local authorities' excellent administrative abilities [38]. To measure the characteristics of sustainable development in tourism research, current research on the influence of tourism categorizes the quantitative indicators by environmental,social, and economical aspects [39]. Besides that, [30] suggested that strategic intervention is required from the governments to attain sustainable tourism development. Government interference by destination management organizations (DMOs) and public bodies can protect these resources, which would support the local economy and the people's welfare. Moreover, tourism promotes the country's economic growth, which assures prosperity and contributes to sustainable development. Based on the theory and literature findings proposed hypothesis is;

*H1*: *Tourism has a statistically significant and positive influence on sustainable development.*

## 2.2. Role of institutional quality in tourism and sustainable development

Institutional quality denotes the quality of institutions that govern laws, government property rights, constitution, and traditions essential for the personal relationship amongst the stakeholders [13]. Previous literature has argued that higher institutional quality can affect the tourists' flow. Also, [40] suggested that institutional quality can play an improtant part in the economy's sustainability in natural resource-abundant countries. Poor governance and political risk hurt tourism activities [41]. Further, [12] investigated the impacts of institutional quality attached to power, socio-economic factors, and political risks on tourism and indicated that institutional quality is a significant determining factor for tourist flows.

Tourist operators and service providers may suspend their activities in the presence of political instability. The military's political involvement obstructs the tourism industry's development due to the lack of security and peace [41, 42]. Current literature is exploring the composite dimensions of political influences on the tourism industry, comprising nations' regional integrity, safety, security, social stability, institutions, and peace because tourist inflows can be influenced by these factors [43–45]. Moreover, [46] revealed that global tourists are concerned about governmental effectiveness, political stability, regulations, laws, and corruption than voice and accountability. In addition, [9] estimated the asymmetries between all variations in institutional quality and tourism inflows from 1980–2018 by utilizing the balanced panel data of the Asian Pacific region. ARDL model was employed by [13] to observe the positive influence of institutional quality on demand of international tourism in India from 1995–2016. The existence of terrorism is very harmful to the tourism sector growth as it can force travellers to alter their travel destination as personal safety is the priority for everyone. However, the effect of corruption on tourism is mixed. Corruption can create a hazard to tourism sector growth as it affects the country's reputation, or it can raise business transactions by criminal and illegal activities (e.g., prostitution and gambling) [9].

Recent literature has also argued that high institutional quality can affect sustainable development. Further, [14] showed a positive and economically significant impact of good institutional quality on sustainable development. Also, [15] stated that institutional quality factors are needed for productivity growth and sustainability. In addition, [16] also highlighted the significance of institutional governance quality on the sustainability of Malaysia from 1985 to 2015. Besides, [47] established the association amongst heritage conservation, tourism and institutional design as tourism supports to create cash for the heritage's maintenance and tourism depend on the institutional design. Also, [48] used the data from 1996 to 2015 to determine the influence of institutional quality amongst tourism and financial development in Malaysia. Control of corruption and government effectiveness played significant role amongst tourism and financial development. Therefore, it can be assumed that institutional quality is a determining factor of tourism development in a country and contributes to sustainability. Based on the theory and literature findings proposed hypothesis is;

H2: *Institutional quality plays a significant, positive moderating role between tourism and sustainable development.*

## 2.3. Other socio-economic factors affecting sustainable development

Other factors include urbanization, exchange rate, renewable energy per capita, per capita income, and household consumption, affecting the SD.

Urbanization signifies the level of urban population comparative to the overall population as per the United Nations report of 2010 [49]. Urbanization provides opportunities for diversity, proximity, and marketplace competition [50]. Previous studies have shown that urbanization hurts biodiversity and effecting sustainable development. Further, [51] provided a clear understanding of the association amongst tourism, urbanization, and climate change for three popular Thailand destinations (Koh Chang, Koh Mak, and Pattaya). Rapid urbanization for tourism development contributes to economic growth, but it was vulnerable to climate change-related risks and other global environmental burdens. Urbanization increases waste and wastewater pollution, inappropriate land-use planning, and resource overconsumption, so urban governance is required to deliver sustainable tourism. Also, [52] explored the tourism-$CO_2$ emissions-urbanization association to prove that the urbanization rate has a positive association with the development of tourism economics while negatively associated with the

emissions of $CO_2$. Therefore, urbanization for tourism development is contributing negatively to sustainable development.

Addressing and modelling the exchange rate is crucial and related to economic growth and meeting social needs, whereby it is necessary for sustainable development and to resolve the transparency and fluctuations in the exchange rate with empirical evidence. Moreover, [53] stated that the exchange rate level is the best proxy to decide about the travel destination. If any country's currency devaluates, it will raise travel flows to that country as international tourism would become less expensive. In addition, [54] examined the degree to which currency appreciations/depreciations would change the United States (U.S.) mutual tourism trade with Canada, the United Kingdom (U.K.), and Mexico. Tourism is influenced by changes in consumer behavior that is generally determined by exchange rate variations. As variations in the country's exchange rate is related to tourist activities, it is imperative to comprehend the exchange rate problems as it would affect the sustainable development.

Moreover, environmental degradation issues are increasing in developing countries. These issues are increasing due to non-renewable energy consumption. Thus, [6] explained the importance of environmental sustainability indicators in the transitional selective tourism destination and [55] stated that the tourism industry is a non-negligible contributor to worldwide $CO_2$ emissions by utilizing provincial panel data from China's tourism industry from 2005–2016. Also, [3] empirically tested the effect of environmental competitiveness on sustainable tourism growth for 130 destinations between 2009 and 2017. The connection amongst environmental competitiveness and sustainable tourism growth was strong for developed destinations while weak for less developing destinations. In addition, [56] stated that renewable energy sources can decrease $CO_2$ emissions and also guarantee sustainable economic development. Thus, the renewable energy per capita is very important, and contributing to the sustainable development.

Per capita income is essential for every country's sustained growth; therefore, [57] used the annual data from 1971 to 2011 for time-serial analysis to measure the impact of Inflation Rate (INF) and Per Capita Income (INC) on sustainability (ANS) and showed the existence of a short-run and long-run relationship amongst these variables. Furthermore, [28] utilized panel data from 1990–2014 for 12 Asian countries to determine the factors that determine sustainable development. Their results indicated a positive impact of per capita income on sustainable development and a negative impact of the inflation rate on sustainable development.

Household consumption is the market value of all goods and services bought by households. It also comprises payments and dues to governments to attain licenses and permits. Household consumption is an essential determinant of sustainability; hence [44] stated that the country's rise in uncertainty might cause a postponement/cancellation of travel plans because of security, safety, and social stability. People would be unwilling to travel out of the country. Households' consumption will decrease if there is uncertainty in an economy. A greater level of uncertainty in any country affects outbound tourism, household consumption, and sustainable development.

## 3. Research methodology

### 3.1. Data and measurement

This study aims to determine the impact of tourism on sustainable development based on 64 BRI countries and data collected from 2003 to 2018 while considering the moderating effect of institutional quality. Other factors include the exchange rate, per capita income, household consumption, urbanization, and renewable energy per capita. Sample countries list has been added in Appendix. The dependent variable sustainable development was measured by an

index of adjusted net saving (ANS), which the World Bank provided. The ANS estimation starts with gross saving equivalent to total income and total consumption's difference [28]. This proxy of sustainable development followed was suggested by [14, 28, 57]. The Independent variable tourism data source is World Bank.

The moderating variable institutional quality (IQ) is based on the six indicators of the International Country Risk Guide (ICRG), i.e., corruption, government stability, law and order, investment profile, bureaucracy quality, and democratic accountability. This proxy was followed by [12, 16]. For the robustness check, we used six Worldwide Governance Indicators (WGI). Moreover, [58] constructed these six indicators, including political stability and the absence of violence, control of corruption, regulatory quality, government effectiveness, the rule of law, and voice and accountability. This proxy was followed by [13, 14, 46, 48].

Moreover, other factors include urbanization, exchange rate, renewable energy per capita income, and household consumptions. As per United Nations (UN), different countries have different urbanization criteria due to national differences in urban characteristics from rural areas. Urbanization signifies the degree to which the urban population is growing [49]. Moreover, [53] stated that the exchange rate level is the best proxy to determine the destination's cost, and the real exchange rate was considered in this study as used by [54, 59]. Local currency units per dollar measured the exchange rate, and the source was The International Monetary Fund (IMF). Renewable energy per capita is also an important influencer of destination competitiveness and are positively related to sustainability. Renewable energy per capita data was taken from World Bank. Household consumption was measured by household consumption as a percent of GDP, and the source was The World Bank data. Sustainable development can be affected by per capita income [28]. As supported by economic theory, income is the critical determinant of savings because it increases the levels of savings when income increases. Per capita income was proxied by Gross National Income per Capita (GNIPC) in a constant local currency unit [57]. In a robustness check, inflation was used as an alternate household consumption variable in this study. Inflation and household consumption were often being perceived as to be correlated. Therefore, the inflation rate was included as one of the explanatory variables as it would affect ANS. Consumer Price Index (CPI) was adopted as a proxy for the inflation rate. The percent change in the CPI measured inflation, and the source is the World Bank data. This proxy is used by [57] and [28].

## 3.2. Model framework

The conceptual framework of the study in Fig 1 is demonstrated below.

Solow's neoclassical model was used to examine the influence of tourism on sustainable development (SD) concerning the BRI region's IQ. The total factor productivity, also known as Solow residual or technological factor, is an imperative parameter. From the studies of [28, 57, 60], economic growth and extended version of SD implemented Cobb–Douglas function, which has been represented as;

$$Yt = At \lceil Kt \rceil \alpha \lceil Lt \rceil \beta \lceil Rt \rceil \gamma \tag{1}$$

The extended version of the SD neoclassical model of Solow by adopted Cobb–Douglas function, which has been represented as;

$$Sy/y = \Phi + \alpha (dy * y*) - \beta n + \gamma \omega + p + \varphi t \tag{2}$$

The dependent variable SD' proxy "ANS" can be supposed as a function of tourism (TUR) and institutional quality composite (IQ). Also, control factors are urbanization (UP), exchange rate (EXR), per capita Income (INC), Inflation (INF), and household consumption (HHC).

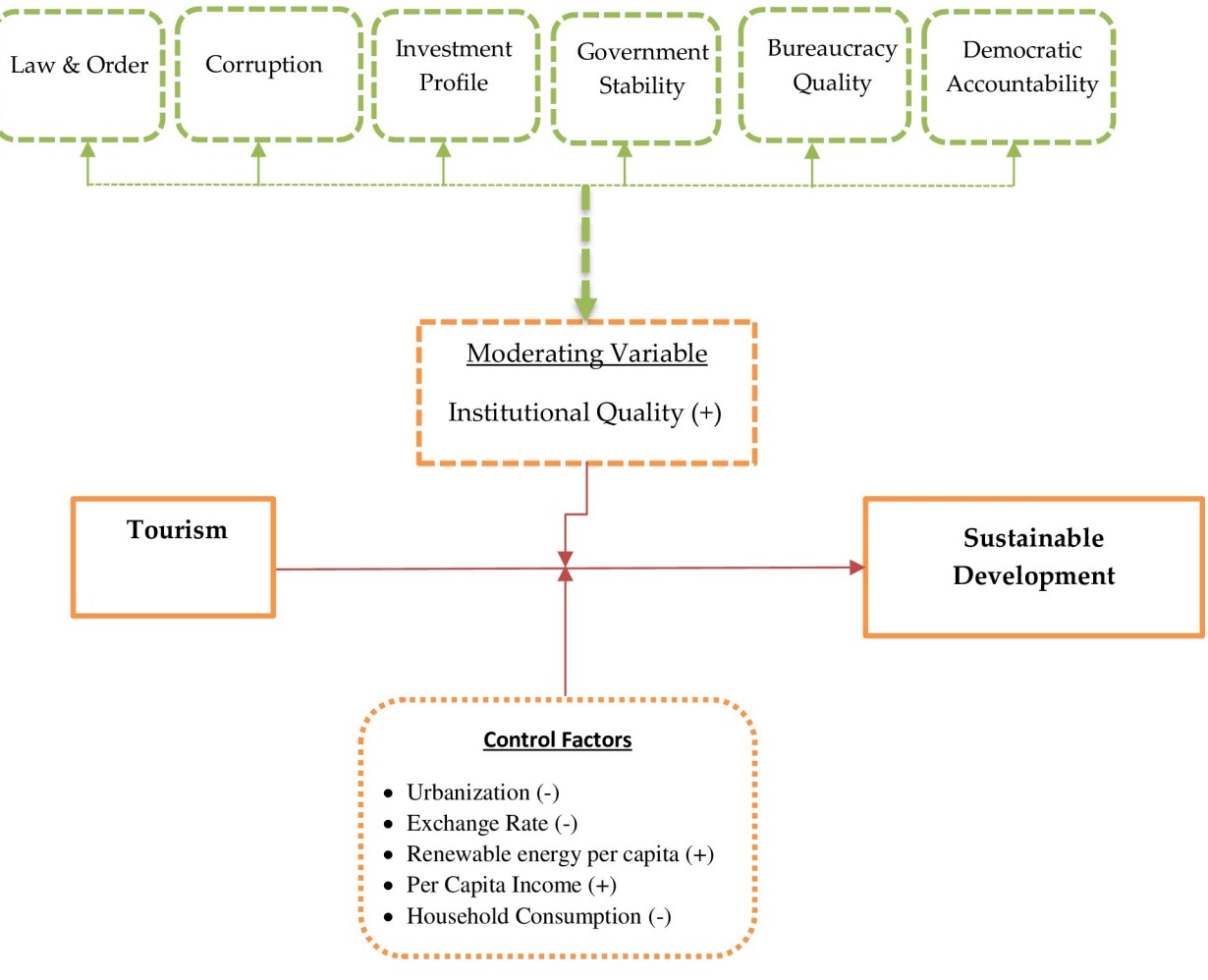

**Fig 1. Conceptual framework.**

Therefore, the model relationship can be expressed as follows;

$$SD\,(ANSR) = \int\,(TUR,\ UP,\ EQ,\ EXR,\ INC,\ HC) \qquad (3)$$

$$SD\,(ANSR) = \int\,(TUR,\ IQ,\ T*IQ,\ UP,\ ER,\ INC,\ HC) \qquad (4)$$

## 3.3. Empirical model of two-step system GMM

GMM technique has been used as it is suitable for our dataset of 64 countries and 16 years (2003–2018). The two-step system GMM technique best examines measurement errors, over-identifying restrictions, auto-correlation in the panel dataset, and endogeneity issues [46, 61]. This technique's vital criterion is that number of cross-sections N must be greater than the period. It is based on static and dynamic models. Hansen test was employed to enhance further investigation by monitoring the over-identifying restrictions. AR 1 p-value value must be less than 0.05, while AR2 p-value must be greater than 0.05.

For the robustness check, we have used alternate variables. The benefit of this technique is that it serves as robustness for each other to observe the consistency of the impact of tourism

variable on sustainable development. Based on the theoretical framework, details explained in Eqs (2) and (3) static and dynamic models are as follows;

**1. Direct channel.**  A static model for measuring the tourism impact on sustainable development;

$$SD_{i,t} = \beta_0 + \beta_1(TUR)_{i,t} + \beta_2(UP)_{i,t} + \beta_3(EQ)_{i,t} + \beta_4(EXR)_{i,t} + \beta_5(INC)i, t + \beta_6(HHC)i, t$$
$$+ \varphi + \mu_i, \tag{5}$$

The dynamic model direct for measuring the influence of the tourism on sustainable development;

$$SD_{i,t} = \beta_0 + \beta_1(SD)_{i,t-1} + \beta_2(TUR)_{i,t} + \beta_3(UP)_{i,t} + \beta_4(EQ)_{i,t} + \beta_5(EXR)_{i,t} + \beta_6(INC)i, t$$
$$+ \beta_7(HHC)_{i,t} + \varphi + \mu_{i,t} \tag{6}$$

**2. Indirect channel.**  Moderating relationship of IQ amongst the tourism and SD by using static and dynamic models has been stated below;

The static model is;

$$SD_{i,t} = \beta_0 + \beta_1(TUR)_{i,t} + \beta_2(IQ)_{i,t} + \beta_3(TUR^*IQ)_{i,t} + \beta_4(UP)_{i,t} + \beta_5(EQ)_{i,t} + \beta_6(EXR)_{i,t}$$
$$+ \beta_7(INC)i, t + \beta_8(HHC)i, t + \varphi + \mu_{i,} \tag{7}$$

The dynamic model is;

$$SD_{i,t} = \beta_0 + \beta_1(SD)_{i,t-1} + \beta_2(TUR)_{i,t} + \beta_3(IQ)_{i,t} + \beta_4(TUR^*IQ)_{i,t} + +\beta_5(UP)_{i,t} + \beta_7(EQ)_{i,t}$$
$$+ \beta_7(EXR)_{i,t} + \beta_8(INC)i, t + \beta_9(HHC)_{i,t} + \varphi + \mu_{i,t} \tag{8}$$

## 4. Results and discussion

### 4.1. Descriptive summary

Table 1 exhibits the descriptive statistics in terms of the standard deviation, mean, minimum and maximum values, and observation count for the selected dependent, independent and control variables. Table 1 shows that a sample of 64 countries has 1024 observations from 2003 to 2018. The descriptive statistics show that all values are within the range.

Table 2 displays the variance inflation factor (VIF). We have calculated the VIF for independent variable tourism and control variables to confirm that there is no multicollinearity in our sample. If the value of VIF would be higher than five for any variable, then it indicates that

**Table 1. Descriptive statistics.**

| Variables | Obs | Mean | Std.Dev. | Min | Max |
|---|---|---|---|---|---|
| Sustainable Development (SD) | 1024 | 10.752 | 11.768 | -15.86 | 38.201 |
| tourism | 1024 | 58.50 | 89.80 | 26300 | 5.300 |
| IQ(ICRG) | 1024 | .004 | 1.002 | -2.81 | 2.582 |
| Tourism*IQ | 1024 | 46.40 | 88.40 | -0.049 | 3.791 |
| Urbanization | 1024 | 55.377 | 21.994 | 16.116 | 100 |
| Renewable Energy Per Capita (EQ) | 1024 | 5.663 | 7.242 | .05 | 39.332 |
| Exchange Rate | 1024 | 749.206 | 2637.302 | .28 | 16302.25 |
| Per Capita Income | 1024 | 3.56 | 3.977 | -8.444 | 14.875 |
| Household | 1024 | 3.801 | 4.713 | -8.892 | 18.834 |
| Inflation | 1024 | 5.216 | 4.928 | -1.3 | 26.8 |

**Table 2. Variance inflation factor.**

| Variables (Dependent: SD) | VIF | 1/VIF |
|---|---|---|
| tourism | 1.09 | 0.916 |
| IQ(ICRG) | 1.77 | 0.564 |
| Tourism*IQ | 1.76 | 0.567 |
| Urbanization | 2.11 | 0.474 |
| Renewable Energy Per Capita(EQ) | 1.96 | 0.510 |
| Exchange Rate | 1.09 | 0.942 |
| Per Capita Income | 1.61 | 0.621 |
| Household | 1.55 | 0.644 |
| Inflation | 1.16 | 0.865 |
| **Mean VIF** | **1.561** | . |

a specific variable has multicollinearity issues [62]. All VIF values in Table 2 are less than five, so our data is free from multicollinearity.

Table 3 displays the correlation between the variables. The results of Table 3 depict that tourism, institutional quality (IQ), urbanization, renewable energy per capita (EQ), have a statistically positive and significant relationship, at 14.1%, 20.8%, 16.6%, and 30.9%, respectively, with sustainable development at 1% significance level. Per capita income is also positively correlated with sustainable development at 5% significance level. However, the exchange rate is negatively correlated with sustainable development at 5% significance level. The independent variable tourism interaction terms and the moderator institutional quality, i.e., Tourism*IQ, depict a positive relationship (12.9%) with sustainable development.

## 4.2. Co-integration test

Co-integration of model panels was confirmed by the Westerlund test. The Westerlund test was performed to confirm the long-term association between the variables considered. The results of the Westerlund test have been depicted in Table 4, indicating a long-term co-integrating association between sustainable development and its determinants. This test specified that all panels were co-integrated.

**Table 3. Pairwise correlations matrix.**

| SR. No. | Variables | (1) | (2) | (3) | (4) | (5) | (6) | (7) | (8) | (9) | (10) |
|---|---|---|---|---|---|---|---|---|---|---|---|
| (1) | SD | 1.000 | | | | | | | | | |
| (2) | Tourism | 0.141*** | 1.000 | | | | | | | | |
| (3) | IQ(ICRG) | 0.208*** | 0.049 | 1.000 | | | | | | | |
| (4) | Tourism*IQ | 0.129*** | -0.114*** | 0.517*** | 1.000 | | | | | | |
| (5) | Urbanization | 0.166*** | 0.205*** | 0.230*** | 0.068** | 1.000 | | | | | |
| (6) | EQ | 0.309*** | 0.143*** | 0.182*** | -0.008 | 0.689*** | 1.000 | | | | |
| (7) | Exchange Rate | -0.073** | -0.025 | 0.001 | -0.020 | -0.221*** | -0.161*** | 1.000 | | | |
| (8) | Per Capita Income | 0.068** | 0.039 | -0.054* | 0.002 | -0.229*** | -0.237*** | 0.084*** | 1.000 | | |
| (9) | Household | 0.010 | 0.097*** | -0.099*** | -0.105*** | -0.099*** | -0.106*** | 0.070** | 0.579*** | 1.000 | |
| (10) | Inflation | -0.088*** | -0.048 | -0.250*** | -0.250*** | -0.250*** | -0.176*** | 0.060* | 0.145*** | 0.148*** | 1.000 |

*** p<0.01

** p<0.05

* p<0.1.

**Table 4. Results of panel co-integration test.**

| Westerlund test for co-integration | Statistic | Decision |
|---|---|---|
| Variance ratio | 2.5396*** | Ha: All panels are co-integrated |
| Pedroni test for co-integration | | |
| Modified Phillips-Perron t | 13.8913 | Ha: All panels are co-integrated |
| Phillips-Perron t | -8.9836 | |
| Augmented Dickey-Fuller t | -7.2286 | |

Note: Number of panels = 64, Number of periods = 16.

P-values are

*** p<0.01

** p<0.05

* p<0.1.

## 4.3. Impact of tourisim and socio-economic factors on sustainable development: Direct channel

Table 5 determines the sustainable development estimation with static and dynamic pooled OLS, fixed-effects models, and two-step system GMM. Static pooled OLS in column 1 demonstrates that tourism positively impacts SD with an R-Squared value of 0.138. Besides, renewable energy per capita and per capita income variables positively and significantly influence sustainability. Based on the Wald test and two-step system GMM criteria's, column 5 exhibits the final model of the two-step system GMM by demonstrating that the coefficient of SD lags (dependent variable), i.e., SD is positive (0.330) with a p-value of less than 1%, which indicates the dynamic nature of SD. Furthermore, findings show that tourism's coefficient is significant and positive (0. 351), with a p-value of less than 1%. Renewable energy per capita and per capita income positively impact sustainable development, while urbanization, exchange rate, and household consumption hurt sustainable development at the 1% significance level. It shows that renewable energy per capita and per capita income contributed to sustainability in the BRI region from 2003 to 2018.

Arellano–Bond (AR1) p-value is less than 5%, while the second-order difference's Arellano–Bond (AR2) p-value is more significant than 5%. The Hansen test statistic value is 45.46 with a p-value of 0.111, and the Sargan test value is 482.9. Moreover, the number of instruments value is 59, which is less than our 64 groups, further specify the validity of the two-step system GMM instruments. Moreover, the Wald test Chi-square demonstrated that the model is fit to use as the p-value is less than 1%. Generally, the diagnostic test results show that all assumptions are correct and estimation techniques are accurate and reliable.

## 4.4. Robustness test of direct channel

The validity of the results was confirmed by substituting the alternative variables and re-estimating similar techniques. In the robust check model, the moderating variable IQ (based on indicators of ICRG) was replaced with Worldwide Governance Indicators (WGI), and household consumption was replaced with inflation. The results of Table 6 exhibited the robustness of tourism impact on SD. Model 5 in column 5 of the two-step system GMM specified that the coefficient of SD lags in an alternate index is positive (0.153), with a p-value of less than 1%, which indicated the dynamic nature of SD. The control variable "Inflation" negatively impacted SD with a coefficient (-0.625), with a p-value of less than 1%. The Hansen test

**Table 5. Results of factors affecting sustainable development–Direct channel.**

| Variables | (1) | (2) | (3) | (4) | (5) |
|---|---|---|---|---|---|
| | Static Model | | Dynamic Model | | |
| | Panel Pooled OLS | Panel Fixed Effect | Panel Pooled OLS | Panel Fixed Effect | Final Model of Two-step System GMM |
| | SD | SD | SD | SD | SD |
| Sustainable Development (SD) | | | 0.944*** | 0.731*** | 0.330*** |
| | | | (0.010) | (0.023) | (0.022) |
| Tourism | 0. 136*** | 0. 241** | 0. 127 | 0. 518 | 0. 351*** |
| | (0. 402) | (0. 110) | (0. 122) | (0. 526) | (0. 709) |
| Urbanization | -0.053** | -0.733*** | 0.001 | -0.129* | -0.501*** |
| | (0.022) | (0.241) | (0.007) | (0.075) | (0.096) |
| Renewable energy per capita (EQ) | 0.634*** | 0.253 | 0.063*** | 0.122 | 1.299*** |
| | (0.066) | (0.641) | (0.021) | (0.136) | (0.168) |
| Exchange rate | -0.000 | 0.002*** | -0.000 | 0.001* | -0.001*** |
| | (0.000) | (0.001) | (0.000) | (0.000) | (0.000) |
| Per Capita Income | 0.531*** | 0.345*** | 0.458*** | 0.501*** | 0.389*** |
| | (0.113) | (0.080) | (0.035) | (0.038) | (0.051) |
| Household Consumptions | -0.169* | -0.130*** | -0.343*** | -0.331*** | -0.193*** |
| | (0.092) | (0.047) | (0.029) | (0.030) | (0.032) |
| | (1.978) | (1.457) | (0.592) | (0.629) | (4.700) |
| Constant | 6.272*** | 42.735*** | 0.760 | 8.309** | 0.000 |
| | (1.808) | (12.230) | (0.559) | (3.891) | (0.000) |
| I. Year | Yes | Yes | Yes | Yes | Yes |
| Observations | 1,024 | 1,024 | 960 | 960 | 960 |
| R-squared | 0.138 | 0.145 | 0.924 | 0.603 | |
| AR1 | . | . | . | . | -3.870 |
| AR1 (p-value) | . | . | . | . | 0.0001 |
| AR2 | . | . | . | . | 0.965 |
| AR2 (p-value) | . | . | . | . | 0.335 |
| Sargan Test | . | . | . | . | 482.9 |
| Hansen Test | . | . | . | . | 45.46 |
| Hansen Test (p-value) | . | . | . | . | 0.111 |
| J-stat | . | . | . | . | 59 |
| Wald Test | . | . | . | . | 14381 |
| Wald Test (p-value) | . | . | . | . | 0 |
| No. of Groups | 64 | 64 | 64 | 64 | 64 |

Standard errors in parentheses

*** p<0.01

** p<0.05

* p<0.1.

statistic value is 41.19 with a p-value of 0.155, and the Sargan test value is 277.4, which illustrates the Hansen and Sargan test validity. Overall, results showed that the model is fit to use.

## 4.5. The IQ moderation role in tourism and sustainable development: Indirect channel

Table 7 validated the IQ moderation amongst tourism and SD. The interaction term of Tourism*IQ has been added. Static model 1 of pooled OLS in column 1 showed that tourism and

**Table 6. Robustness check of factors afffacting sustainable development results–direct channel.**

| Variables | (1) | (2) | (3) | (4) | (5) |
|---|---|---|---|---|---|
| | Static Model | | Dynamic Model | | |
| | Panel Pooled OLS | Panel Fixed Effect | Panel Pooled OLS | Panel Fixed Effect | Final Model of Two-step System GMM |
| | SD | SD | SD | SD | SD |
| Sustainable Development | | | 0.936*** | 0.679*** | 0.153*** |
| | | | (0.010) | (0.024) | (0.029) |
| Tourism | 1.307*** | 0. 210** | -0. 298 | 0. 191 | 0. 328** |
| | (0. 399) | (0. 105) | (0. 130) | (0. 560) | (0. 128) |
| Urbanization | -0.064*** | -0.705*** | -0.001 | -0.158** | -0.782*** |
| | (0.023) | (0.231) | (0.007) | (0.080) | (0.202) |
| Renewable energy per capita (EQ) | 0.631*** | 0.300 | 0.066*** | 0.226 | 1.841*** |
| | (0.066) | (0.637) | (0.023) | (0.144) | (0.424) |
| Exchange rate | -0.000 | 0.002*** | -0.000 | 0.001* | -0.001** |
| | (0.000) | (0.001) | (0.000) | (0.000) | (0.000) |
| Per Capita Income | 0.437*** | 0.257*** | 0.249*** | 0.301*** | 0.256*** |
| | (0.098) | (0.076) | (0.033) | (0.036) | (0.080) |
| Inflation | -0.209*** | -0.152*** | -0.025 | -0.050 | -0.625*** |
| | (0.079) | (0.056) | (0.027) | (0.035) | (0.079) |
| Constant | 7.516*** | 41.675*** | 0.272 | 9.183** | 43.281*** |
| | (1.894) | (11.704) | (0.630) | (4.144) | (8.695) |
| I. Year | Yes | Yes | Yes | Yes | Yes |
| Observations | 1,024 | 1,024 | 960 | 960 | 960 |
| R-squared | 0.141 | 0.146 | 0.913 | 0.549 | |
| AR1 | . | . | . | . | -4.087 |
| AR1 (p-value) | . | . | . | . | 0.0000 |
| AR2 | . | . | . | . | -0.218 |
| AR2 (p-value) | . | . | . | . | 0.827 |
| Sargan Test | . | . | . | . | 277.4 |
| Hansen Test | . | . | . | . | 41.19 |
| Hansen Test (p-value) | . | . | . | . | 0.155 |
| J-stat | . | . | . | . | 57 |
| Wald Test | . | . | . | . | 351.5 |
| Wald (p-value) | . | . | . | . | 0 |
| No. of Groups | 64 | 64 | 64 | 64 | 64 |

Standard errors in parentheses

*** p<0.01

** p<0.05

* p<0.1.

IQ positively impacted SD. The two-step system GMM results in column 5 showed that SD is positive (0.166), with a p-value of less than 1%, which signifies the dynamic nature of SD. The comparative moderating composite of IQ indicated that the coefficient is positive (4.693), and the coefficient of tourism was also positive (3.717) with a p-value of less than 1%. However, the interaction term of Tourism*IQ specifies that the coefficient is (- 1.507). Furthermore, other determinants, i.e., renewable energy per capita and per capita income, indicated a significant and positive impact on SD, while urbanization, household consumption, and exchange rate negatively influenced SD.

**Table 7. Moderating results of IQ between tourism and sustainable development: Indirect channel.**

| Variables | (1) | (2) | (3) | (4) | (5) |
|---|---|---|---|---|---|
| | Static Model | | Dynamic Model | | |
| | Panel Pooled OLS | Panel Fixed Effect | Panel Pooled OLS | Panel Fixed Effect | Final Model of Two-step System GMM |
| | SD | SD | SD | SD | SD |
| Sustainable Development | | | 0.945*** | 0.730*** | 0.166*** |
| | | | (0.010) | (0.023) | (0.033) |
| Tourism | 1.467*** | 2.397** | 1.041 | 5.018 | 3.717*** |
| | (0. 401) | (0. 108) | (0. 124) | (0. 528) | (0. 780) |
| Institutional Quality–IQ (ICRG) | 1.442*** | 0.792 | 0.117* | 0.452 | 4.693*** |
| | (0.450) | (1.267) | (0.142) | (0.638) | (0.589) |
| Tourism*IQ | 9.938** | -4.610 | -1.621 | -1.708 | - 1.507*** |
| | (0. 504) | (0. 101) | (0. 156) | (0. 493) | (0. 526) |
| Urbanization | -0.073*** | -0.735*** | 0.001 | -0.130* | -0.645*** |
| | (0.022) | (0.242) | (0.007) | (0.075) | (0.116) |
| Renewable Energy Per Capita (EQ) | 0.637*** | 0.241 | 0.060*** | 0.117 | 1.524*** |
| | (0.066) | (0.648) | (0.021) | (0.136) | (0.213) |
| Exchange Rate | -0.000 | 0.002*** | -0.000 | 0.001* | -0.001*** |
| | (0.000) | (0.001) | (0.000) | (0.000) | (0.000) |
| Per Capita Income | 0.484*** | 0.347*** | 0.459*** | 0.502*** | 0.247*** |
| | (0.112) | (0.080) | (0.035) | (0.038) | (0.049) |
| Household Consumptions | -0.103 | -0.130*** | -0.344*** | -0.330*** | -0.072* |
| | (0.091) | (0.047) | (0.029) | (0.030) | (0.040) |
| Constant | 7.146*** | 42.924*** | 0.783 | 8.389** | 0.000 |
| | (1.786) | (12.337) | (0.563) | (3.900) | (0.000) |
| I. Year | Yes | Yes | Yes | Yes | Yes |
| Observations | 1,024 | 1,024 | 960 | 960 | 960 |
| R-squared | 0.169 | 0.146 | 0.924 | 0.603 | |
| AR1 | . | . | . | . | -3.520 |
| AR1 (p-value) | . | . | . | . | 0.000432 |
| AR2 | . | . | . | . | 0.893 |
| AR2 (p-value) | . | . | . | . | 0.372 |
| Sargan Test | . | . | . | . | 322.3 |
| Hansen Test | . | . | . | . | 39.54 |
| Hansen Test (p-value) | . | . | . | . | 0.201 |
| J-stat | . | . | . | . | 59 |
| Wald Test | . | . | . | . | 1513 |
| Wald Test (p-value) | . | . | . | . | 0 |
| No. of Groups | | 64 | | 64 | 64 |

Standard errors in parentheses

*** p<0.01

** p<0.05

* p<0.1.

The estimated indirect channel model indicated the zero autocorrelation and serial correlation in first-order and second-order difference because the AR1 p-value is less than 5% and the AR2 p-value is more significant than 5%, respectively. The Hansen test statistic value is 39.54, with a p-value of 0.201, and the Sargan test value is 322.3, which illustrated the

instrument reliability. The number of instruments is 59, which is less than the number of groups (64) and specifies GMM instruments' validity. The diagnostic test results specify the accuracy and reliability of estimation techniques.

## 4.6. Robustness test of indirect channel

The indirect channel IQ moderation results between tourism and sustainable development in BRI countries are confirmed through a robustness test presented in Table 8. The moderating variable IQ (based on indicators of ICRG) was replaced with Worldwide Governance Indicators (WGI), and household consumption was replaced with inflation. Column 1 of static Pooled OLS shows that tourism significantly impacts SD with p-values less than 1%. Robustness results are reported in column 5. The value of IQ is (5.972), with a p-value less than 1%, which indicates the moderating role of IQ amongst tourism and SD.

Moreover, tourism has a significant and positive coefficient (3.187) with the p-values of less than 1%. The robust interaction term of Tourism*IQ indicates that the coefficient is (-0. 253), with a p-value less than 1%. Renewable energy per capita and per capita income show a significant, positive impact on SD, although urbanization, exchange rate, and inflation show a negative but significant influence on SD, with p-values less than 1%.

AR1 and AR2 values also fulfilled the criteria. The Hansen test statistic value is 38.29, with a p-value of 0.206, and the Sargan test value is 257.4, which indicates the instrument reliability. Furthermore, the numbers of instruments are less than the numbers of groups, which further indorse the validity of GMM instruments.

## 4.7. Discussion of results and summary of key findings

The outcomes of the Westerlund, Pedroni, and Kao tests confirmed that the panel is co-integrated. Correlation analysis results validated that tourism, institutional quality, renewable energy per capita, and per capita income positively affected SD. In contrast, the interaction of Tourism*IQ, urbanization, inflation, and the exchange rate negatively correlates with SD, with a p-value of less than 1%. Both direct and indirect channel outcomes specified the two-step system GMM reliability. The results showed that the SD coefficient was positive with a p-value of less than 1%, which signifies the dynamic nature of sustainable development. Both channels confirm that tourism positively impacts sustainable development in favour of the hypothesis (H1), which indicates that tourism leads to the SD. Our results are in line with [4].

The hypothesis (H2) was also confirmed from the results that institutional quality plays a significant moderating role between tourism and sustainability in BRI countries from 2003–2018. The robustness check also demonstrated positive and significant influence, which authenticates our hypothesis. Our results are in line with [14–16]. Urbanization was found to be a significant negative contributor to BRI countries' sustainability, and our results are in line with [51]. The exchange rate showed a negative influence on sustainable development. Renewable energy per capita has a positive and significant influence on sustainability, and or results are in line with [56]. Moreover, a significant and positive contribution of per capita income was observed to BRI countries' sustainability, and our results are in line with [28]. Household consumption negatively impacted sustainable development, and robustness check variable inflation showed the same results, so our results are in line with [28].

## 5. Conclusion and policy implications

This study addressed the dynamic influence of tourism on sustainable development in 64 BRI countries from 2003 to 2018. This study also pointed out the moderating impact of institutional quality and effects of other determinants such as urbanization, exchange rate, renewable

**Table 8. Robustness check results of moderating impact of IQ between tourism and SD.**

| Variables | (1) | (2) | (3) | (4) | (5) |
|---|---|---|---|---|---|
| | Static Model | | Dynamic Model | | |
| | Panel Pooled OLS | Panel Fixed Effect | Panel Pooled OLS | Panel Fixed Effect | Final Model of Two-step System GMM |
| | SD | SD | SD | SD | SD |
| Sustainable Development | | | 0.934*** | 0.678*** | 0.088*** |
| | | | (0.010) | (0.024) | (0.031) |
| Tourism | 1.427*** | 2.080** | -3.461 | 1.53 | 3.187*** |
| | (0. 039) | (0. 103) | (0.013) | (0.056) | (0.091) |
| Institutional Quality (IQ) (WGI) | 1.420*** | 0.690 | 0.216 | 0.669 | 5.972*** |
| | (0.452) | (1.226) | (0.152) | (0.679) | (0.582) |
| Tourism*IQ | 9.498* | -3.80 | -5.30 | -4.478 | -0. 253*** |
| | (0. 509) | (0. 946) | (0. 016) | (0.080) | (0. 591) |
| Urbanization | -0.078*** | -0.707*** | -0.003 | -0.160** | -0.635*** |
| | (0.022) | (0.233) | (0.008) | (0.080) | (0.141) |
| Renewable energy per capita (EQ) | 0.636*** | 0.289 | 0.066*** | 0.216 | 1.457*** |
| | (0.066) | (0.644) | (0.023) | (0.145) | (0.280) |
| Exchange Rate | -0.000 | 0.002*** | -0.000 | 0.001* | -0.001** |
| | (0.000) | (0.001) | (0.000) | (0.000) | (0.000) |
| Per Capital Income | 0.425*** | 0.259*** | 0.249*** | 0.304*** | 0.179*** |
| | (0.096) | (0.075) | (0.033) | (0.036) | (0.047) |
| Inflation | -0.088 | -0.149** | -0.017 | -0.047 | -0.232*** |
| | (0.081) | (0.057) | (0.028) | (0.035) | (0.064) |
| Constant | 7.579*** | 41.839*** | 0.338 | 9.380** | 35.245*** |
| | (1.870) | (11.823) | (0.632) | (4.152) | (6.466) |
| I. Year | Yes | Yes | Yes | Yes | Yes |
| Observations | 1,024 | 1,024 | 960 | 960 | 960 |
| R-squared | 0.169 | 0.147 | 0.913 | 0.550 | |
| AR1 | . | . | . | . | -3.350 |
| AR1 (p-value) | . | . | . | . | 0.0008 |
| AR2 | . | . | . | . | 0.745 |
| AR2 (p-value) | . | . | . | . | 0.456 |
| Sargan Test | . | . | . | . | 257.4 |
| Hansen Test | . | . | . | . | 38.29 |
| Hansen (p-value) | . | . | . | . | 0.206 |
| J-stat. | . | . | . | . | 58 |
| Wald Test | . | . | . | . | 838.5 |
| Wald (p-value) | . | . | . | . | 0 |
| No. of Groups | | 64 | | 64 | 64 |

Standard errors in parentheses

*** p<0.01

** p<0.05

* p<0.1.

energy per capita, per capita income, inflation, and household consumption on sustainability. Furthermore, outcomes were validated using the relevant indicators for institutional quality (such as WGI) and household consumption (such as inflation). Direct and indirect channel results indicated that the two-step system GMM is the best technique for this research. Direct

and indirect channel findings of the two-step system GMM validated the significant positive dynamic nature of sustainable development and its association with tourism and other determinants in the BRI region. This study theoretically contributes that BRI countries are on a sustainable path, and tourism contributes positively to the BRI region's sustainability to achieve the SDGs agenda. Per capita income and renewable energy per capita also promote sustainability. Renewable energy per capita is an essential indicator of sustainable tourism development. Renewable energy per capita is a significant contributor to destination competitiveness and are positively associated with destination countries' financial development.

The interaction amongst Tourism*IQ was found negative, suggesting that institutional quality should be improved between BRI countries to endorse and improve sustainable tourism development effectually. Governments should make stable political situations, especially in those countries that depend on tourism for economic growth. Such reforms will be beneficial to obtain additional advantages from the tourism industry. Political risk can be reduced by improving mutual diplomatic connections, security, and safety. Exchange rate stabilization is beneficial for the tourism industry and sustainable development. Changes can influence the tourism industry in customer behaviour, which also depends on exchange rate fluctuations. The countries must impose few restrictions regarding exchange rate policies and maintain stable and predictable exchange rates as it affects tourism inflows. Exchange rate fluctuation is affecting the travel and tourism industry in BRI countries. Rapid urbanization because of tourism growth is the key driver of ecological changes as it is increasing waste and wastewater pollution, inappropriate land-use planning, and resource overconsumption. Urban governance, e.g., proper laws and regulations, effective administrative and political procedures, and robust local institutional capacity, are obligatory for urban development to deliver sustainability. As trade-offs exist amongst economic and ecological standards as infrastructure development inevitably comprises substantial risks for biodiversity. Therefore, infrastructure development should be done cautiously to guarantee negligible negative influences on biodiversity, and stakeholders must plan substitute solutions to avoid the negative impacts. Deliberative decision-making tools are well established for multivariate problems, reducing the impacts associated with infrastructure development.

The findings of current research have important policy implications for balanced and sustainable growth. This study's practical implication refers to encouraging policymakers and institutions to contribute to tourism activities to increase sustainability. Government intervention is required to foster land planning, regulate and legislate, create incentives to investment, protect historical and folk heritage, and endorse tourism for sustainable development. The current study also emphasizes and identifies gaps in the domain and body of knowledge. The limitation of the study is the incomplete data for some BRI countries. Secondly, genuine savings do not include R&D expenditures. This study explored the impact of tourism on sustainable development while considering the institutional quality. Future studies should consider technological factors like e-government determinants that impact sustainable development.

## Supporting information

**S1 Appendix. Sample countries of belt and road.**
(DOCX)

**S1 Dataset.**
(XLSX)

## Acknowledgments

Authors acknowldge the Associate Professor Ningyu Qian for his assistance, support and encourgement during the whole process.

## Author Contributions

**Conceptualization:** Huma Iftikhar, Chen Pinglu, Saif Ullah, Atta Ullah.

**Data curation:** Atta Ullah.

**Formal analysis:** Atta Ullah.

**Funding acquisition:** Chen Pinglu.

**Investigation:** Huma Iftikhar.

**Methodology:** Chen Pinglu, Saif Ullah, Atta Ullah.

**Project administration:** Chen Pinglu.

**Resources:** Chen Pinglu.

**Software:** Atta Ullah.

**Supervision:** Chen Pinglu.

**Validation:** Atta Ullah.

**Visualization:** Huma Iftikhar, Chen Pinglu, Saif Ullah.

**Writing – original draft:** Huma Iftikhar.

**Writing – review & editing:** Chen Pinglu, Saif Ullah, Atta Ullah.

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
