## [Decision Letter · Decision Letter 0]

22 Sep 2021

PONE-D-21-26959Impact of Tourism on Sustainable Development in BRI Countries: The Moderating Role of Institutional QualityPLOS ONE

Dear Dr. Pinglu,

Thank you for submitting your manuscript to PLOS ONE. After careful consideration, we feel that it has merit but does not fully meet PLOS ONE’s publication criteria as it currently stands. Therefore, we invite you to submit a revised version of the manuscript that addresses the points raised during the review process.

We look forward to receiving your revised manuscript.

Kind regards,

Jun Yang

Academic Editor

PLOS ONE

Journal Requirements:

3. Please ensure that you refer to Figure 1 in your text as, if accepted, production will need this reference to link the reader to the figure.

Additional Editor Comments:

Reviewer 1

You can refer to the following articles to further improve the quality of your articles：

Impact of climate-related changes to the timing of autumn foliage colouration on tourism in Japan. Tourism Management, 70,262–272.

Impact of climate change on hiking: quantitative evidence through big data mining.Current Issues in Tourism. https://doi.org/10.1080/13683500.2020.1858037.

Effects of rural revitalization on rural tourism. Journal of Hospitality and Tourism Management (2021),https://doi.org/10.1016/j.jhtm.2021.02.008.

Study on the Impact of High-speed Railway Opening on China's Accessibility Pattern and Spatial Equality.Sustainability 2018,10,2943. doi:10.3390/su10082943.

The influence of high-speed rail on ice–snow tourism in northeastern China. Tourism Management（2020）, doi:10.1016/j.tourman.2019.104070.

Reviewer 2

This study investigated the influence of tourism on sustainable development in BRI Countries, institutional quality as a moderating variable. The research has certain practical significance. But there are still some aspects to be improved.

(1) There are too many contents of literature review. Try to select the most relevant articles to comment on the research topic of this paper, and there is no need to pursue comprehensive.

(2) As the key of this paper, the description of conceptual framework is insufficient.

(3) To what extent does tourism affect sustainable development should be clarified in the abstract. Otherwise, the impact of tourism on sustainable development seems to be obvious, even without this study.

(4) There is little description for Table 1, 2, 3, 4. Any table in the paper should be necessary, meaningful and clearly described.

(5) Compared with previous studies, what is the innovation of this paper? Further clarification is required.

Reviewers' comments:

Reviewer's Responses to Questions

**Comments to the Author**

1. Is the manuscript technically sound, and do the data support the conclusions?

Reviewer #1: Yes

Reviewer #2: Yes

2. Has the statistical analysis been performed appropriately and rigorously? 

Reviewer #1: Yes

Reviewer #2: Yes

3. Have the authors made all data underlying the findings in their manuscript fully available?

Reviewer #1: Yes

Reviewer #2: Yes

4. Is the manuscript presented in an intelligible fashion and written in standard English?

Reviewer #1: Yes

Reviewer #2: Yes

5. Review Comments to the Author

Reviewer #1: You can refer to the following articles to further improve the quality of your articles

Effects of rural revitalization on rural tourism. Journal of Hospitality and Tourism Management (2021),https://doi.org/10.1016/j.jhtm.2021.02.008.

Study on the Impact of High-speed Railway Opening on China's Accessibility Pattern and Spatial Equality.Sustainability 2018,10,2943. doi:10.3390/su10082943.

The influence of high-speed rail on ice–snow tourism in northeastern China. Tourism Management（2020）, doi:10.1016/j.tourman.2019.104070.

Reviewer #2: This study investigated the influence of tourism on sustainable development in BRI Countries, institutional quality as a moderating variable. The research has certain practical significance. But there are still some aspects to be improved.

(1) There are too many contents of literature review. Try to select the most relevant articles to comment on the research topic of this paper, and there is no need to pursue comprehensive.

(2) As the key of this paper, the description of conceptual framework is insufficient.

(3) To what extent does tourism affect sustainable development should be clarified in the abstract. Otherwise, the impact of tourism on sustainable development seems to be obvious, even without this study.

(4) There is little description for Table 1, 2, 3, 4. Any table in the paper should be necessary, meaningful and clearly described.

(5) Compared with previous studies, what is the innovation of this paper? Further clarification is required.

6. PLOS authors have the option to publish the peer review history of their article (what does this mean?). If published, this will include your full peer review and any attached files.

Reviewer #1: No

Reviewer #2: No

---

## [Author Response · Author response to Decision Letter 0]

3 Jan 2022

We are also extremely grateful for the valuable suggestions and constructive comments from the reviewers, which have been instrumental in strengthening the manuscript significantly. We have carefully addressed the reviewer each comment and do so sincerely hope that the manuscript meets the expectations. Please find improvements in file 'Revised Manuscript with Track Changes' in Yellow Highlighted texts. Moreover, after comments revision, overall proofreading was done by authors. Thank you very much for your precious time. 

Reviewer: 1. 

Comment:

You can refer to the following articles to further improve the quality of your articles：

Impact of climate-related changes to the timing of autumn foliage colouration on tourism in Japan. Tourism Management, 70,262–272.

Impact of climate change on hiking: quantitative evidence through big data mining. Current Issues in Tourism. https://doi.org/10.1080/13683500.2020.1858037.

Effects of rural revitalization on rural tourism. Journal of Hospitality and Tourism Management (2021),https://doi.org/10.1016/j.jhtm.2021.02.008.

Study on the Impact of High-speed Railway Opening on China's Accessibility Pattern and Spatial Equality. Sustainability 2018,10,2943. doi:10.3390/su10082943.

The influence of high-speed rail on ice–snow tourism in northeastern China. Tourism Management（2020）, doi:10.1016/j.tourman.2019.104070.

Response:

We are thankful for the valuable suggestions. Recommended articles have been cited in the manuscript to improve the quality of article.

Changes incorporated in “Introduction” section

Page 5, lines 1 to 9.

Previous researchers have investigated the factors that affect tourism, e.g. high-speed railway increases the approachability of ice-snow tourist destinations and contributing to the global tourism industry (Jin, Yang, Wang, & Liu, 2020; Yang, Guo, Li, & Huang, 2018). Rural revitalization promotes rural tourism activities and contributes to sustainable rural development in developing countries (Yang et al., 2021). While climate change is a critical factor that influences the tourism industry negatively. Humidity, sunshine duration and temperature affect hiking participation nonlinearly (Liu, Yang, Zhou, & Wang, 2020). Climate-related changes also impact tourism as it reduce the duration of leaf colouration in Japan, which causes a delay in autumn foliage colouration change (Liu, Cheng, Jiang, & Huang, 2019). 

Reviewer: 2. 

Comment Summary:

This study investigated the influence of tourism on sustainable development in BRI Countries, institutional quality as a moderating variable. The research has certain practical significance. But there are still some aspects to be improved.

Response:

Thank you for the encouragement. We are incredibly thankful for the valuable suggestions and constructive comments to improve the manuscript. We have carefully tried to address each comment and do so sincerely hope that the manuscript meets your expectations. 

Comment 1:

There are too many contents of literature review. Try to select the most relevant articles to comment on the research topic of this paper, and there is no need to pursue comprehensive.

Response 1:

The literature review part is improved as suggested. We have tried our best to select most relevant articles. Word count of literature review has been reduced from 2039 to 1799. 

Changes incorporated in “Literature Review” section. 

Page 5 to 10.

Comment 2:

As the key of this paper, the description of conceptual framework is insufficient.

Response 2:

Further description has been added on your kind suggestion.

Changes incorporated in “Research Methodology” section. 

Page 12.

Model Framework 

The conceptual framework of the study is demonstrated below;

[Insert Figure 1. Conceptual Framework Model]

Solow's neoclassical model was used to examine the influence of tourism on sustainable development (SD) concerning the BRI region's IQ. The total factor productivity, also known as Solow residual or technological factor, is an imperative parameter. From the studies of (Koirala & Pradhan, 2019; Pardi et al., 2015; Ullah, Pinglu, Ullah, & Hashmi, 2021b), economic growth and extended version of SD implemented Cobb–Douglas function, which has been represented as; 

𝑌𝑡 = 𝐴𝑡 ⌈𝐾𝑡⌉α ⌈𝐿𝑡⌉β ⌈𝑅𝑡⌉γ (1) 

The extended version of the SD neoclassical model of Solow by adopted Cobb–Douglas function, which has been represented as;

Sy/y=Φ+α (dy* y*)− βn + γω + p +𝜑𝑡 (2)

The dependent variable SD' proxy "ANS" can be supposed as a function of tourism (TUR) and institutional quality composite (IQ). Also, control factors are urbanization (UP), exchange rate (EXR), per capita Income (INC), Inflation (INF), and household consumption (HHC). Therefore, the model relationship can be expressed as follows;

SD (ANSR) = ∫ (TUR, UP, EQ, EXR, INC, HC) (3)

SD (ANSR) = ∫ (TUR, IQ, T*IQ, UP, ER, INC, HC) (4)

Comment 3:

To what extent does tourism affect sustainable development should be clarified in the abstract? Otherwise, the impact of tourism on sustainable development seems to be obvious, even without this study.

Response 3:

Improved and incorporated the modification as suggested. 

The current research findings have valuable contributions to academics as it offers novel insights about the 0. 351% influence of tourism on sustainable development at significance level of 1%, and it proposes valued suggestions to policymakers concerning tourism development strategies.

The outcomes of the two-step system GMM confirmed the statistically significant and positive dynamic nature of sustainable development and its relationship with tourism and other determinants at a significance level of 1% for BRI countries. Institutional quality enhanced the 4.693% sustainability path to achieve the sustainable development goals (SDGs) agenda with regionally interconnected countries at significance level of 1%. Renewable energy per capita and per capita income played a significant and positive role, while the exchange rate, household consumption, and urbanization negatively influenced sustainable development. 

Comment 4:

There is little description for Table 1, 2, 3, 4. Any table in the paper should be necessary, meaningful and clearly described.

Response 4:

Improved, further description has been added for Table 1,2,3,4. 

Changes incorporated in “Results and Discussion” section.

Page 14 to 17.

4.1 Descriptive Summary 

Table 1 exhibits the descriptive statistics in terms of the standard deviation, mean, minimum and maximum values, and observation count for the selected dependent, independent and control variables. Table 1 shows that a sample of 64 countries has 1024 observations from 2003 to 2018. The descriptive statistics show that all values are within the range.

Table 2 displays the variance inflation factor (VIF). We have calculated the VIF for independent variable tourism and control variables to confirm that there is no multicollinearity in our sample. If the value of VIF would be higher than five for any variable, then it indicates that a specific variable has multicollinearity issues (Mahmood, Han, Ali, Mubeen, & Shahzad, 2019). All VIF values in Table 2 are less than five, so our data is free from multicollinearity.

Table 3 displays the correlation between the variables. The results of table 3 depict that tourism, institutional quality (IQ), urbanization, renewable energy per capita (EQ), have a statistically positive and significant relationship, at 14.1%, 20.8%, 16.6%, and 30.9%, respectively, with sustainable development at 1% significance level. Per capita income is also positively correlated with sustainable development at 5% significance level. However, the exchange rate is negatively correlated with sustainable development at 5% significance level. The independent variable tourism interaction terms and the moderator institutional quality, i.e., Tourism*IQ, depict a positive relationship (12.9%) with sustainable development.

4.2 Co-integration Test 

Co-integration of model panels was confirmed by the Westerlund test. The Westerlund test was performed to confirm the long-term association between the variables considered. The results of the Westerlund test have been depicted in Table 4, indicating a long-term co-integrating association between sustainable development and its determinants. This test specified that all panels were co-integrated. 

Comment 5:

Compared with previous studies, what is the innovation of this paper? Further clarification is required.

Response 5:

As mentioned in introduction part (Page#5) that previous study (Ibanescu et al., 2018) on this topic was done for one country only. Authors mentioned that data was poor because it does not allow contrasts to other states. This study has been done for 64 countries of BRI region. As most of countries of BRI region have natural beauty and economies of these countries depends on tourism. No previous study was done for this region.

Secondly, this study is the further extension of (Ullah, Pinglu, Ullah, & Hashmi, 2021a)’ work. Authors suggested the researchers to work on these variables in future. So, we measured the impact of tourism on sustainable development and considered institutional quality as a moderator. We have mentioned this study in our introduction part as a motivation for our paper.

Changes incorporated in “Introduction” section.

Page 5.

(Ibanescu et al., 2018) investigated how tourism development has affected the sustainable development for Romania only. This study was based on Romanian statistics data at the LAU-2 level, and this sustainable development data was poor as per the author because it does not permit contrasts to other states. Similarly, the role of institutional quality is crucial for both tourism and sustainable development, so institutional quality was added as a moderator as suggested by (Ullah, Pinglu, Ullah, & Hashmi, 2021a). As compared to other countries, the BRI region has natural beauty so it is considered a preferred destination by worldwide tourists. Tourism is very important for the growth and revenues of BRI region as it comprises many low and middle-income nations. However, the influence of tourism on sustainable development was not explored for the BRI region, and institutional quality was not considered as a moderator between tourism and sustainable development in previous studies. Therefore, this research focuses on how tourism contributes to sustainable development with the moderating role of institutional quality by using country-level aggregated BRI countries' data from 2003 to 2018. This approach also facilitated the confirmation of whether urbanization, renewable energy per capita, exchange rate, per capita income, inflation, and household consumption would influence the sustainable development for the BRI region, which would be a vital contribution in the literature.

---

## [Decision Letter · Decision Letter 1]

26 Jan 2022

Impact of Tourism on Sustainable Development in BRI Countries: The Moderating Role of Institutional Quality

PONE-D-21-26959R1

Dear Dr. Ullah,

We’re pleased to inform you that your manuscript has been judged scientifically suitable for publication and will be formally accepted for publication once it meets all outstanding technical requirements.

Kind regards,

Jun Yang

Academic Editor

PLOS ONE

Additional Editor Comments (optional):

Accept

Reviewers' comments:

Reviewer's Responses to Questions

**Comments to the Author**

1. If the authors have adequately addressed your comments raised in a previous round of review and you feel that this manuscript is now acceptable for publication, you may indicate that here to bypass the “Comments to the Author” section, enter your conflict of interest statement in the “Confidential to Editor” section, and submit your "Accept" recommendation.

Reviewer #1: All comments have been addressed

Reviewer #2: All comments have been addressed

2. Is the manuscript technically sound, and do the data support the conclusions?

Reviewer #1: Yes

Reviewer #2: Yes

3. Has the statistical analysis been performed appropriately and rigorously? 

Reviewer #1: I Don't Know

Reviewer #2: Yes

4. Have the authors made all data underlying the findings in their manuscript fully available?

Reviewer #1: No

Reviewer #2: Yes

5. Is the manuscript presented in an intelligible fashion and written in standard English?

Reviewer #1: Yes

Reviewer #2: Yes

6. Review Comments to the Author

Reviewer #1: (No Response)

Reviewer #2: The authors have addressed all the concerns, I have no more comments. I recommend the manuscript to be accepted for publication.

7. PLOS authors have the option to publish the peer review history of their article (what does this mean?). If published, this will include your full peer review and any attached files.

Reviewer #1: No

Reviewer #2: No

---

## [Editor Report · Acceptance letter]

8 Apr 2022

PONE-D-21-26959R1 

Impact of tourism on sustainable development in BRI countries: the moderating role of institutional quality 

Dear Dr. Ullah:

I'm pleased to inform you that your manuscript has been deemed suitable for publication in PLOS ONE. Congratulations! Your manuscript is now with our production department. 

Kind regards, 

on behalf of

Dr. Jun Yang 

Academic Editor

PLOS ONE